# Regional Variability and Driving Forces behind Forest Fires in Sweden

**Reinis Cimdins** [1,*], **Andrey Krasovskiy** [2] and **Florian Kraxner** [2]

1  School of Forest Sciences, University of Eastern Finland, 80101 Joensuu, Finland
2  Agriculture Forestry and Ecosystem Services (AFE) Group, Biodiversity and Natural Resources (BNR) Program, International Institute for Applied Systems Analysis (IIASA), Schlossplatz 1, A-2361 Laxenburg, Austria
*  Correspondence: reinic@uef.fi

**Abstract:** Extreme forest fires have been a historic concern in the forests of Canada, the Russian Federation, and the USA, and are now an increasing threat in boreal Europe, where recent fire events in 2014 and 2018 drew attention to Sweden. Our study objective was to understand the vulnerability of Swedish forests to fire by spatially analyzing historical burned areas, and to link fire events with weather, landscape, and fire-related socioeconomic factors. We developed an extensive database of $1 \times 1$ km$^2$ homogenous grids, where monthly burned areas were derived from the MODIS FireCCI51 dataset. The database consists of various socio-economic, topographic-, forest-, and weather-related remote sensing products. To include new factors in the IIASA's FLAM model, we developed a random forest model to assess the spatial probabilities of burned areas. Due to Sweden's geographical diversity, fire dynamics vary between six biogeographical zones. Therefore, the model was applied to each zone separately. As an outcome, we obtained probabilities of burned areas in the forests across Sweden and observed burned areas were well captured by the model. The result accuracy differs with respect to zone; the area under the curve (AUC) was 0.875 and 0.94 for zones with few fires, but above 0.95 for zones with a higher number of fire events. Feature importance analysis and their variability across Sweden provide valuable information to understand the reasons behind forest fires. The Fine Fuel Moisture Code, population and road densities, slope and aspect, and forest stand volume were found to be among the key fire-related factors in Sweden. Our modeling approach can be extended to hotspot mapping in other boreal regions and thus is highly policy-relevant. Visualization of our results is available in the Google Earth Engine Application.

**Keywords:** random forest; forest fire; FFMC; AUC; MODIS

## 1. Introduction

Forest fires, often triggered by human activities and greatly exacerbated by climate change, cause widespread destruction and are therefore an important global issue [1–3]. They contribute to greenhouse gas emissions, air pollution and associated health problems, as well as social and economic disruption for people in fire-prone regions-problems targeted by the Sustainable Development Goals [4]. While vegetation burning plays an important ecological role in some ecosystems [5], climate change leads to longer fire seasons in many parts of the world with an associated trend towards increasing extent, frequency, and severity of fires, as well as the global occurrence of extreme (mega) fire events [6–8]. However, burned areas are decreasing in some regions of the world due to land use change, which could result in a net-negative global trend of historical burned areas over the period 1998–2015 [9].

Forest fires are a familiar issue in Europe [10,11], where most of the burned areas have historically been found in the Mediterranean region [11–13]. Forest fire models project an increase in expected burned area and associated emissions under future climate change scenarios in Europe [12,13]. Fire is an increasing threat in the European Boreal forests

affected by climate change due to lengthened fire seasons [14]. Recent fire events in 2014 and 2018 have drawn attention to a need for more fire-related research in Sweden [15,16].

There is a variety of global models trying to simulate forest fire dynamics, including CLM, JSBACH–SPITFIRE, LPJ–GUESS–SPITFIRE, and ORCHIDEE–SPITFIRE. However, they are largely unable to capture inter-annual variations in historical burned areas [17]. Having strong biophysical components, these models can represent feedback between forest fires and vegetation dynamics driven by climate change, but they lack representation of interconnections between the vegetation and human activities due to the complexity of these processes. This creates a research gap, because to explain current fire-regime changes and assess probabilities of fire, climate change needs to be considered in combination with other drivers [18].

The wildfire climate impacts and adaptation model (FLAM) is able to capture inter-annual dynamics of burned areas by calibrating the spatially explicit suppression efficiency [19,20]. However, the parameter representing suppression efficiency is rather technical and needs interpretation in terms of fire-related factors. Hot-spot mapping is a way to understand factors behind fire ignitions and suppression efficiency. However, identifying hot spots in Europe is still a challenging task when modeling at relatively high spatial resolution [21]. In this paper, we concentrate on hot-spot mapping for one European country and perform analysis at $1 \times 1$ km$^2$ resolution.

Our study objective was to understand the vulnerability of Swedish forests to fire by spatially analyzing historical burned areas and linking them with weather, landscape, topography and fire-related socioeconomic factors. In this way, our research contributes to the development of a wildfire model for Sweden [22]. Our study expands its analysis to include an extreme fire event in 2018, has a larger number of predictors and uses machine learning techniques [23] to find key variables and their interconnections. Our study considers climatic controls [24] in combination with socio-economic and geographic factors, leading to an integrated hot-spot mapping [25–27]. We developed an extensive database of $1 \times 1$ km$^2$ homogenous grids, where monthly areas burned in forests were derived from the MODIS FireCCI51 dataset [28]. Spatial factors, including campsites, lakes, and roads, topographic features including aspect, slope, and mean elevation, population density, forest management intensity, and forest stand volume, were collected from various sources and preprocessed. Monthly values of the Fine Fuel Moisture Code (FFMC) of the Canadian Forest Fire Weather Index (FWI) [25,26] over the period 2011–2018 were calculated from daily weather data with the FLAM model [20]. To include new factors into FLAM, we developed a random forest model [27] to assess the spatial probabilities of burned areas [29–32]. The model contained fifteen features representing various aspects of wildfire behavior and was based on available remote sensing products. Model performance was assessed using the Area Under the Curve (AUC) method, which is a commonly used approach for classification model evaluation [33,34]. Feature importance analysis and their variability across Sweden provide valuable information to understand the factors behind forest fires. FFMC, population and road densities, slope and aspect, and forest stand volume were found to be among the key fire-related factors in Sweden. Visualization of our results is available in the Google Earth Engine Application (available at: https://reiniscimdins.users.earthengine.app/view/swedenmaps (accessed on 10 October 2022)).

## 2. Materials and Methods

### 2.1. Study Area

Wildfire modeling was performed on forests in Sweden; the country has an elongated shape with a latitudinal difference of 14 degrees (~1500 km). The presence of Scandinavian mountains results in climatic zone, biome, economic activity and population pattern diversity in Sweden. This variability impacts wildfire regimes and potential fire occurrences across the regions of Sweden. Therefore, in our study Sweden was divided into six vegetation zones, within which modeling was performed separately (Figure 1). During the study period, Sweden suffered from two severe fire years—2014 and 2018. In 2014, a

fire occurred in southern Sweden (Västmanland), 150 km NW from Stockholm, burning ~14,000 ha, causing 1000 evacuations and one fatality [35]. A second heatwave-induced fire took place in 2018. When fires were distributed throughout the country, the total burned area was ~ 24,300 ha, with the largest burned areas in Gävleborg and Jämtland counties, with ~8500 ha in each of the regions [36]. These fires spread mostly in low populated areas and therefore only a few villages were evacuated. Due to the extraordinary extent of the fire, fire suppression was assisted and supported by numerous European countries [35].

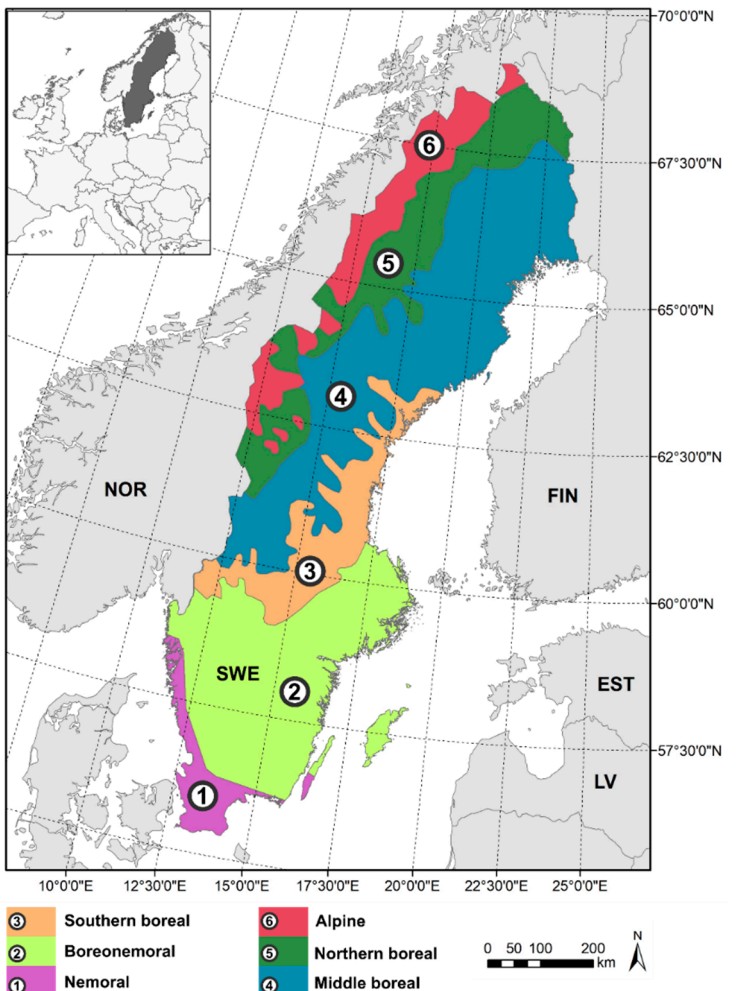

**Figure 1.** Study area (Sweden) and vegetation zones used for modeling.

## 2.2. Data Preparation

Our approach follows methodology previously applied to other countries, such as Italy [29], Portugal [30] and Serbia [34], aiming to search for the spatial factors behind forest fires, their importance and impact on probabilities of fire ignition and burned areas. To understand the linkage between historical fire activity and weather, landscape, topography and fire-related socioeconomic factors, we established a $1 \times 1$ km$^2$ gridded geospatial database which was generated using the ETRS89 LAEA Europe metric coordinate system (EPSG:3035) and contained fifteen fire-related parameters. Variables and data flows are illustrated in Figure 2.

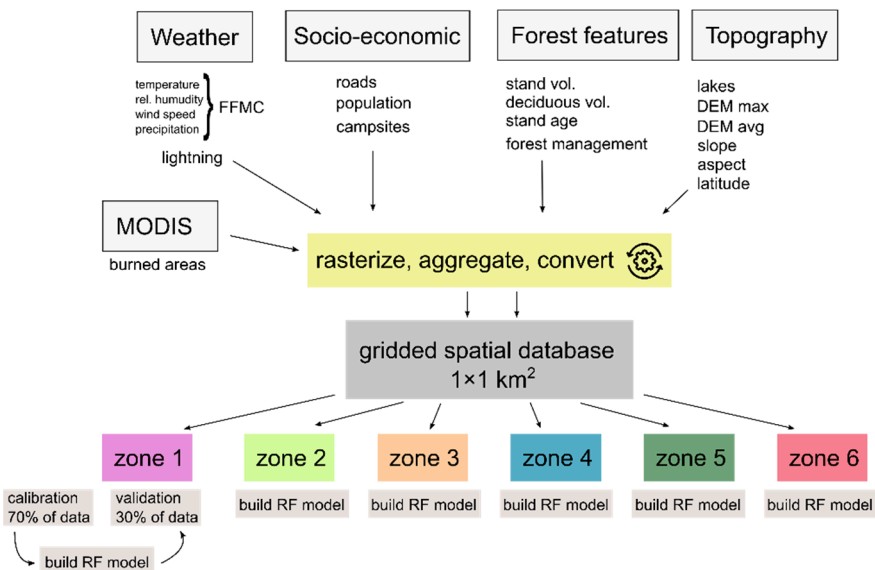

**Figure 2.** Data processing workflow.

Daily computations of the Fine Fuel Moisture Code (FFMC) were performed using the weather module of the FLAM model [20] based on daily HARMONIE (CRUHAR v.3.2) weather data [31]. Outputs provided at the $0.125 \times 0.125°$ grid in the WGS84 coordination system were resampled to the $1 \times 1$ km$^2$ grid and aggregated to a monthly time step. Another weather-related feature we included in the database was lightning, which can be considered as a natural source of fire ignition that also plays a significant role in northern Scandinavia [37]. We obtained data on lightning strikes across Sweden from the Swedish Meteorological and Hydrological Institute (SMHI). Lightning strikes were processed for each pixel for every month from 2011 to 2018.

To construct a database, we used the following remote sensing products. Wildfire data was obtained from the global remote-sensing-based fire product MODIS FireCCI_5_1 [32]. This dataset has a spatial resolution of 250 m and monthly temporal resolution. MODIS data consist of three components: day of fire detection, the confidence level of detection and land cover of burned area, based on the land cover product developed by [38]. This study focused on forest fires; therefore, we used the MODIS land cover codes and selected only those fires which occurred in the forests of Sweden. MODIS data was aggregated to the $1 \times 1$ km$^2$ resolution, where each pixel contained a binary value representing the presence or absence of fire.

Probabilities of forest fire ignition and spread are conditional on the fuel availability and its characteristics. It is assumed that older conifer forests in boreal conditions are more vulnerable to fire as compared to younger stands with a larger deciduous tree ratio in the forest composition [37]. To quantify and represent the fuel available for burning in the model, we collected information about total forest stand volume (m$^3$), deciduous tree volume (m$^3$) and stand age (years). These variables were aggregated to the database grid from the Swedish national forest inventory data, representing the state of forests in 2010 [39] at $25 \times 25$ m$^2$ resolution. Another forest descriptive parameter that was included in the analysis represented forest management intensity. This information was aggregated and extracted from a $100 \times 100$ m$^2$ global forest management map [40], where intensity is approximated to values from 0 to 1. Moreover, the spatial database contains information about topographic features such as digital elevation model (DEM) maximum, average values, mean slope and mean aspect values, which were calculated from the $25 \times 25$ m$^2$ resolution Japan Aerospace Exploration Agency (JAXA) space product [41] and averaged to $1 \times 1$ km$^2$ pixels. These topographic features have an important effect on vegetation type and moisture content [42]. For example, fire can spread faster on upward slopes, where vegetation gets more preheated during wildfire [43]. The population and road densities

are important factors related to forest fire occurrence and suppression [44,45]. We made use of municipality level population density from the $1 \times 1$ km$^2$ resolution NASA dataset, representing the year 2015 [46]. Road density was obtained from the Open-street map database [47]. The road vector layer was rasterized to $20 \times 20$ m$^2$, and sums of road pixels for each $1 \times 1$ km$^2$ grid cell were used as indicator of road density. The forest fire ignition, suppression and isolation are also dependent on waterbody presence in the landscape. We used the CORINE land cover dataset to calculate waterbody ratio (between 0 and 1), representing water coverage in each $1 \times 1$ km$^2$ grid cell [48]. During 2006–2010, 80% of forest fires with a known source in northern Europe were started by humans [49]. For this reason, we generated the shelter and bonfire place density layer, which was based on voluntary mapped locations [50], as a proxy to possible ignitions related to irresponsible handling of fire during recreational activities in nature. As Sweden has a significant latitudinal gradient (Figure 1), we used each pixel center latitude coordinate as one of the explanatory variables, thereby summarizing weather and socioeconomic patterns, varying along the north–south latitude. A summary of all variables is provided in Table 1.

**Table 1.** Modeling variable summary.

| Nr. | Variable | Temporal Resolution | Original Spatial Resolution, m | Source |
|---|---|---|---|---|
| 1 | Fine Fuel Moisture Code | Monthly | 1000 | [31] |
| 2 | lightning | Monthly | 1000 | [51] |
| 3 | MODIS FireCCI_5_1 | Monthly | 250 | [32] |
| 4 | stand volume | Static | | |
| 5 | deciduous tree volume | Static | 25 | [39] |
| 6 | stand age | Static | | |
| 7 | forest management | Static | 100 | [40] |
| 8 | elevation maximum | Static | | |
| 9 | elevation average | Static | | |
| 10 | mean slope | Static | 25 | [41] |
| 11 | mean aspect | Static | | |
| 12 | population density | Static | 1000 | [46] |
| 13 | road density | Static | Vector (lines) | [47] |
| 14 | lake density | Static | 250 | [48] |
| 15 | campsites | Static | Vector (points) | [50] |

### 2.3. Model Setup

In this study we developed a random forest fire occurrence model based on multiple weather, landscape and anthropogenic factors. Random forest is a widely used method for wildfire modeling that has been applied to various regions of the world, e.g., Europe [52], China [53], Canada [54] and the US [55]. The random forest algorithm is a multipurpose regression and classification method which was introduced in 2001 by Breiman [56].

The algorithm builds the decision trees using data subsets and assesses their outputs [27]. The modeling database was divided into six parts according to the vegetation zones (Figure 1) and the model was applied individually in each zone to better understand the factor importance differences across Sweden, as well as reduce the model computation time.

The random forest model was built using the R package *caret* [57] and it was set up ignoring any collinearity relationships between the variables, meaning that all fifteen fire predictors were considered. Each zone had calibration and validation datasets with 70% and 30% of randomly selected rows, respectively. Every model was tuned using four options to define the most appropriate number of variables (mtry 2, 4, 8, 15) at each of the random forest splits. In all zones, we used mtry value 8 because it produced the highest accuracy. Another parameter which influences the model accuracy is number of trees (ntree); for our models, ntree was 70 and it was the same in all the zones and mtry trials. The model was validated using three cross-validation rounds.

## 3. Results

Our analysis shows a seasonal pattern of forest fires in Sweden, with the majority of events taking place from May to September (Figure 3), when most of the recorded burned pixels correspond to historical Fine Fuel Moisture Code (FFMC) values between 80 and 85. The Fine Fuel Moisture Code is a quantification of moisture content of dead fuel components and other litter. This value indicates the relative ignition and potential flammability of fine fuels [58]. At the same time, in spring and early autumn FFMC values are below 80, which is explained by high precipitation during these months (Figure 3a). Zones 3 and 4 contained the largest burned areas over the study period, which could result from favorable fuel and fire propagation conditions and lower suppression efficiency. While zones 1 and 2 are densely populated with intensive agriculture management, the northern zones (5 and 6) have sparse population with limited fuel potential, especially in the mountainous areas (Figures 1 and 3b).

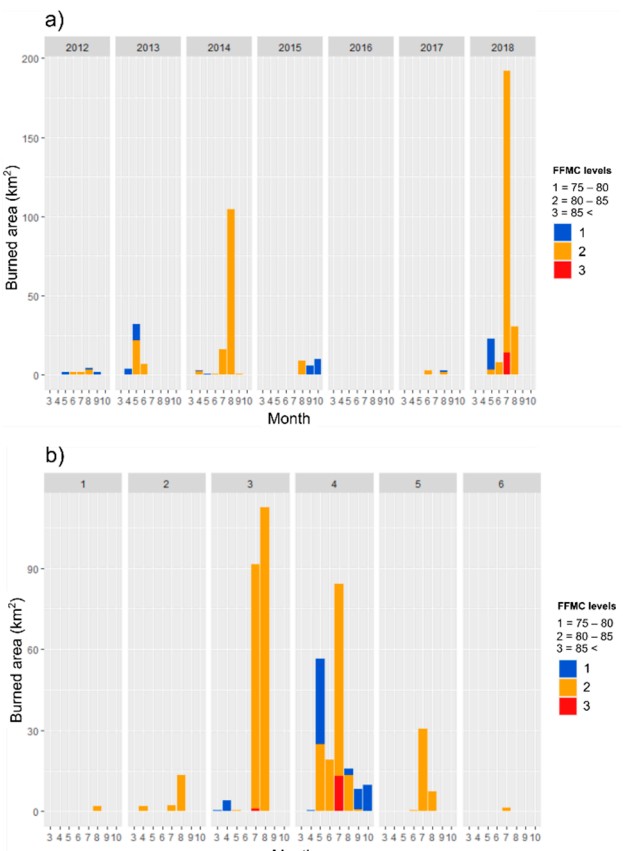

**Figure 3.** Burned area in Sweden divided by year (**a**) and vegetation zone (**b**).

Due to Sweden's geographical diversity, the fire dynamics in Sweden vary between six biogeographical zones. Therefore, the model was applied to each zone separately. Observed burned areas were well captured by the model, providing useful information

about the distribution of fire risk across Sweden (Figure 3). Its performance was determined using the Area Under the Curve (AUC) method and approach, a commonly used approach for classification model evaluation [33,34]. The model prediction accuracy was tested using validation datasets and all the data (validation and calibration). Model accuracy differed across the zones, but in all six vegetation areas validation datasets produced a prediction accuracy above 85%. Moreover, when all information was taken into consideration, AUC was at least 95% (Table 2).

**Table 2.** AUC performance for each zone, considering the validation dataset and all the data.

|  | AUC | |
|---|---|---|
|  | **Validation** | **All** |
| Zone 1 | 0.875 | 0.961 |
| Zone 2 | 0.999 | 0.999 |
| Zone 3 | 0.990 | 0.997 |
| Zone 4 | 0.903 | 0.951 |
| Zone 5 | 0.950 | 0.983 |
| Zone 6 | 1.000 | 1.000 |

The fire prediction results were visualized in Google Earth Engine Application (Figure 4). This application gives an opportunity to explore the study site and use the intuitive Google Maps interface and layer comparison tools to visually assess the model performance with the reference burned area (MODIS). Maps with forest fire predictions show model capabilities to distinguish the main fire areas in the northern part of Sweden (Figure 4). Predictions match with the MODIS data, having some scattered overestimated probability around the reference fire pixels.

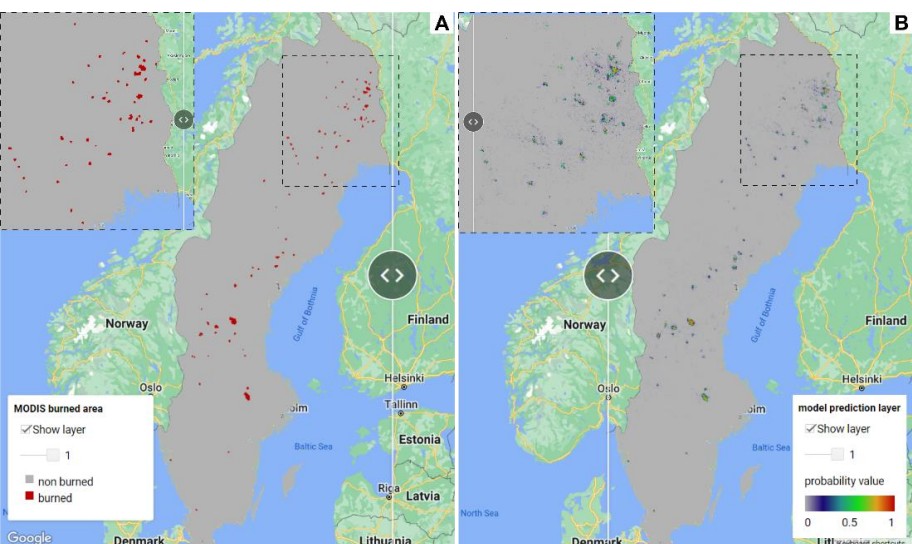

**Figure 4.** MODIS burned area (**A**) and random forest probability values (**B**) from our Google Earth Engine app.

Modeling outcomes were evaluated using confusion matrix-based threshold analysis. We produced an equal-size interval thresholds from 0.1 to 0.5 with a step of 0.1 to better understand the sensitivity and specificity tradeoff concept of probability classification (Figure 5). Despite the relevant differences in terms of the vegetation zone sizes and fire frequencies, the results reveal a consistent and general trend where increased classification threshold values have a positive relation with false negative results, but true positive and false positive values decrease with increasing threshold values. Threshold levels should be determined considering the rationale of the decision-makers by providing risks that

come with the commission and omission error occurrence. These trendlines provide useful information to consider the tradeoffs that come with the classification tasks.

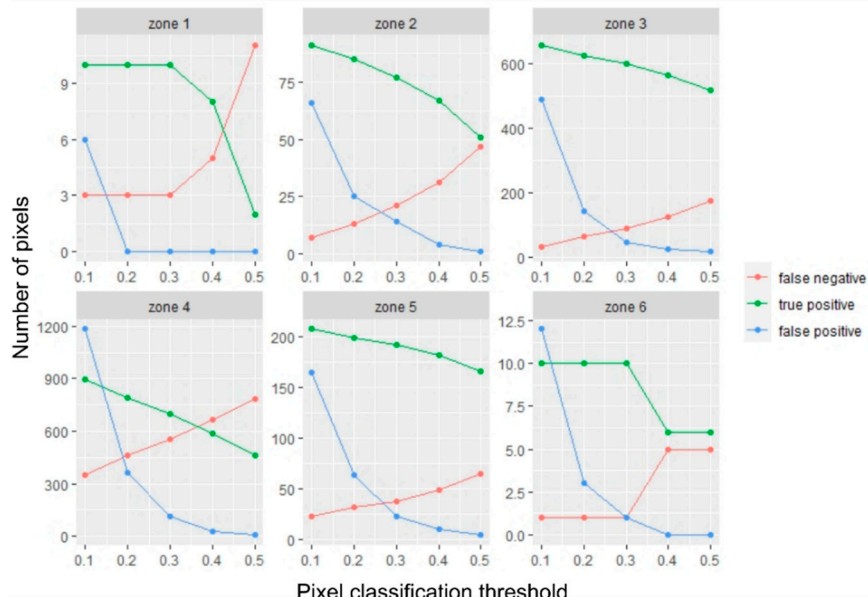

**Figure 5.** Impacts of the random forest prediction threshold on the sensitivity and specificity results.

*Analysis of Importance of Random Forest Factors*

Feature importance analysis and their variability across Sweden provide valuable information to better understand and evaluate factors behind forest fires. Each factor has a different impact on forest fires in each vegetation zone; therefore, we prepared a summary with the variable explanatory value and its corresponding importance rank (Table 3). Latitude and FFMC values were the dominant explanatory variables in all six vegetation zones, except zone 1 where FFMC ranked 9th in terms of importance. We averaged the variable importance metric in all six zones. This measure reports the Mean Square Error percentage increase (%IncMSE) if a certain variable is excluded from a set of predictors—the greater the %IncMSE value, the higher the contribution of the variable to the model's prediction. FFMC and Latitude were the most powerful drivers for the prediction model, having average importance values of 78.2 and 86.0, respectively. Lightning strikes, another weather-related variable, only ranked at the 11th position overall, with an average %IncMSE of 12%. Socio-economic components such as population and road densities could be considered as the second most important category related to forest fire occurrence in the Swedish landscape. Population density had an average ranking of 5.2 and 46.4%IncMSE. One of the predictors—the number of campsites in each of the spatial database grid cells—was expected to contribute to more advanced quantification of possible threats associated with recreation, with an assumption that recreational activities contribute to irresponsible fire use. Nonetheless, this explanatory variable had the least meaningful effect in this study (see Table 3), not supporting our hypothesis. Topographic features, such as mean elevation value, slope and aspect, provided meaningful impacts on the model performance across the study area. Aspect, with a mean importance position of 9.8 and %IncMSE of 20.6%, was the least important topographic feature. In direct contrast is the most important topographical feature, average pixel elevation value (DEM avg), with a mean position of 5.7 and %IncMSE of 35.2%. Forest stand properties were expected to be an important variable to describe the potentially flammable environment; however, forest-linked characteristics such as stand volume, deciduous stand, stand age and management intensity in general had the least influence on predictions. Two of the most important forest characteristics in this model were forest stand volume and forest management information.

Forest stand data exclusion from the model resulted in an average prediction MSE drop of 30.6%, whereas forest management exclusion produced a 24.2% drop (Table 3).

**Table 3.** Random forest variable importance in each vegetation zone.

| Variable | Zone 1 | | Zone 2 | | Zone 3 | | Zone 4 | | Zone 5 | | Zone 6 | | Average | |
| --- | --- | --- | --- | --- | --- | --- | --- | --- | --- | --- | --- | --- | --- | --- |
| | Rank | Value | Rank | Value | Rank | Value | Rank | Value | Rank | Value | Rank | Value | Rank | Value |
| FFMC | 9 | 31.3 | 2 | 84.6 | 1 | 100.0 | 1 | 100.0 | 1 | 100.0 | 1 | 100.0 | 2.5 | 86.0 |
| lightning | 7 | 35.6 | 7 | 20.7 | 15 | 0.0 | 14 | 7.8 | 15 | 0.0 | 9 | 10.9 | 11.2 | 12.5 |
| latitude | 1 | 100.0 | 1 | 100.0 | 2 | 44.7 | 2 | 81.0 | 2 | 72.4 | 2 | 70.8 | 1.7 | 78.2 |
| lakes | 5 | 52.9 | 12 | 6.9 | 9 | 18.9 | 10 | 25.8 | 11 | 12.2 | 9 | 10.9 | 9.3 | 21.3 |
| roads | 3 | 73.9 | 9 | 16.5 | 10 | 18.1 | 7 | 34.0 | 10 | 14.6 | 4 | 53.2 | 7.2 | 35.1 |
| stand vol. | 4 | 55.5 | 13 | 4.2 | 5 | 25.2 | 6 | 34.2 | 3 | 36.1 | 8 | 28.3 | 6.5 | 30.6 |
| DEM max | 6 | 45.3 | 5 | 29.0 | 7 | 23.0 | 9 | 29.7 | 5 | 31.3 | 7 | 40.8 | 6.5 | 33.2 |
| DEM avg | 8 | 32.9 | 6 | 26.3 | 6 | 24.0 | 5 | 34.8 | 6 | 28.7 | 3 | 64.2 | 5.7 | 35.2 |
| population | 2 | 91.3 | 4 | 32.6 | 3 | 39.8 | 3 | 60.3 | 13 | 11.1 | 6 | 43.2 | 5.2 | 46.4 |
| deciduous vol. | 15 | 0.0 | 14 | 4.1 | 13 | 14.5 | 13 | 16.8 | 12 | 11.2 | 9 | 10.9 | 12.7 | 9.6 |
| stand age | 10 | 28.9 | 8 | 20.7 | 12 | 15.4 | 12 | 20.2 | 9 | 17.7 | 15 | 0.0 | 11.0 | 17.2 |
| slope | 14 | 12.9 | 3 | 44.1 | 8 | 19.1 | 11 | 24.1 | 8 | 18.6 | 5 | 46.5 | 8.2 | 27.6 |
| aspect | 13 | 21.8 | 11 | 12.9 | 11 | 17.9 | 4 | 41.5 | 7 | 20.0 | 13 | 9.5 | 9.8 | 20.6 |
| campsites | 11 | 26.9 | 15 | 0.0 | 14 | 3.9 | 15 | 0.0 | 14 | 2.8 | 9 | 10.9 | 13.0 | 7.4 |
| forest management | 11 | 26.9 | 10 | 14.3 | 4 | 32.5 | 8 | 30.1 | 4 | 34.2 | 14 | 7.0 | 8.5 | 24.2 |

## 4. Discussion

The strong influence of FFMC on fire probability values means that climate change will most likely affect the fire season by prolonging it in the spring and autumn directions. Venäläinen and Aalto [59] published extensive research about the effects of climate change and forest management on forest fire occurrence in Fennoscandia. They stressed the uncertainty that comes with long-term projections and concluded that high fire seasons in Sweden will remain occasional. Climate change might influence forest conditions, which would likely impact fire occurrence. New climatic conditions could realistically increase the drought-, tree uprooting-, bark beetle- and deadwood-related fire risks.

Numerous scientists have concluded that fire occurrence is dependent on population [60] and road densities [11], but density increment can result in mixed effects [22]. More inhabitants and increased accessibility might result in increasing recreational activities, sometimes leading to irresponsible attitudes with fire. Knorr et al. [61] reported that fire occurrence has a positive correlation with population density only when it ranges from 0 to 0.1 people/km$^2$. Increased population density usually comes with improved infrastructure which increases the forest management intensity and quality, raises people's awareness about the processes in the forest and considerably improves fire suppression activities in case of raging fire. One of the predictors in our random forest model was the presence of campsites, but this variable had the smallest explanatory power with the lowest average rank and %IncMSE value. This could be explained by non-existing relationship or randomness, but on the other hand, forests next to the campsites could be more intensively managed, thereby reducing fuel availability and ignition potential.

Scientists in Finland [59] and Austria [62] proposed that 10% and 15%, respectively, of all fires are started by lightning strikes. Similarly, our study indicates the importance of lightning, although it is hard to compare our results with previous studies, because our study indicates general lightning importance in each of the random forest models. Here, we note that monthly dynamics of lightning strikes across Sweden (see Figure A1) shows a positive correlation with extreme burned areas in 2014 and 2018, peaking in July. Research focused on lightning ignitions would require analysis at a daily time step.

Forest operation, i.e., harvesting and site preparation, leads to an additional risk for fire ignition and propagation in the Swedish forests. The Västmanland fire in 2014 was caused by rocky soil preparation using heavy mechanical machinery. These soil precautions sometimes take place soon after the clear cut when felling leftovers and tree debris might lose their moisture content and become a potentially highly flammable material [59]. The

study explanatory variable list could be extended with the latest forest activity type and data, which in combination with forest soil or geology information might be an interesting asset, because the combination of site condition and management activities might explain fire occurrence patterns in Sweden. A limiting factor for historical fire data analysis is the consistency of the data collection methodology. Harmonized data is a key aspect to avoid misrepresented conclusions and biased decisions.

High variability in explanatory variables and a large dataset size have led to extensive computation times and required the usage of high-performance hardware. Therefore, optimization of algorithms, perhaps using cloud computing, would be important for future research. Future research could provide worthy additions if we could evaluate how accurate a model would perform if only one country-wise model were calibrated to predict fire occurrence. Fire model performance might get worse if we tested how the model would predict the fire occurrence using years outside the calibration time range (2011–2018). Another investigation could be to test the performance of the model when using separate fire event datasets for the validation and calibration pixels. The performed analysis contributes to the implementation of additional fire-related factors in process-based models, e.g., FLAM [20], to optimize conditional probabilities of ignition, suppression, and fire spread. This helps to capture the spatial and intertemporal variability of areas burned [19].

There are several policy implications of our results. First, the hot-spot mapping could be an important tool for identification of areas which are potentially vulnerable to forest fires. This could help policy makers in their management decisions, e.g., increasing suppression efficiency by optimizing infrastructure and logistics of fire fighters, or preventing fires by putting warning signs in the forest areas. Second, the threshold analysis presented in Figure 5 allows policy makers to consider the trade-offs between false alarms and missed forest fires due to no alarm. Policy makers could consider costs associated with these alternatives and adjust threshold values accordingly. Finally, our results show that climate is one of the main drivers of forest fires in Sweden. Therefore, policy makers at a global scale are recommended to integrate climate change mitigation with guided development.

## 5. Conclusions

This study was developed to understand the weather, topographic, forest stand, socioeconomic and weather factor interlinkages, and the ability to predict historically burned areas using a random forest model. Our study demonstrates fire hotspot mapping at a high resolution. Furthermore, it provides an advanced methodology for using fifteen openly available data sources to achieve high model accuracy in each of the six vegetation zones in Sweden. The Area Under the Curve (AUC) values changed from zone to zone, not undercutting a value of 0.875. The modeled fire probabilities were evaluated using thresholds, thereby producing more information on tradeoffs between the classification's specificity and sensitivity concepts. This provides policy makers the opportunity to improve decision making over missed fire detection and false alerts. The Fine Fuel Moisture Code (FFMC) and latitudinal pixel value have been identified as the main driving forces for forest fires in all vegetation zones. This research provides additional contributions to the existing forest fire knowledge about the situation in boreal forests of Northern Europe. Finally, our results support international analyses that, irrespective of changes in management, it is evident that climate change is very likely to increase the frequency and impact of wildland fires in the coming decades, also in Scandinavia.

**Author Contributions:** Conceptualization, R.C. and A.K.; methodology, R.C. and A.K.; software, R.C.; validation, R.C., A.K. and F.K.; formal analysis, R.C.; investigation, R.C.; resources, R.C. and A.K.; data curation, R.C. and A.K.; writing—original draft preparation, R.C., A.K. and F.K.; writing—review and editing, R.C., A.K. and F.K.; visualization, R.C.; supervision, A.K. and F.K.; project administration, R.C., A.K. and F.K.; funding acquisition, A.K. and F.K. All authors have read and agreed to the published version of the manuscript.

**Funding:** This research was funded by the Short-Term Scientific Mission (STSM) of the COST Action (grant no. CA18135) "Fire in the Earth System: Science & Society" (FireLinks), supported by COST (European Cooperation in Science and Technology). This research was funded by the project "Integrated Future Wildfire Hot Spot Mapping for Austria (Austria Fire Futures)" number C265157, funded by the Climate and Energy Fund and carried out within the framework of the Austrian Climate Research Program (ACRP).

**Data Availability Statement:** Visualization of modeling results is available at: https://reiniscimdins.users.earthengine.app/view/swedenmaps (accessed on 10 October 2022).

**Acknowledgments:** The study was developed during Erasmus Mundus Joint Master's Degree Program (EMJMD)-European Forestry (MSc EF) summer internship 2021 which was hosted by the International Institute for Applied Systems Analysis (IIASA). A preliminary draft of this study has greatly benefited from expert feedback at the Third International Conference on Fire Behavior and Risk [63]. We are grateful to colleague Shelby Corning for proofreading (written) assistance. Furthermore, we would like to acknowledge the International Boreal Forest Research Association (IBFRA, www.ibfra.org (accessed on 10 October 2022)).

**Conflicts of Interest:** The authors declare no conflict of interest.

## Appendix A

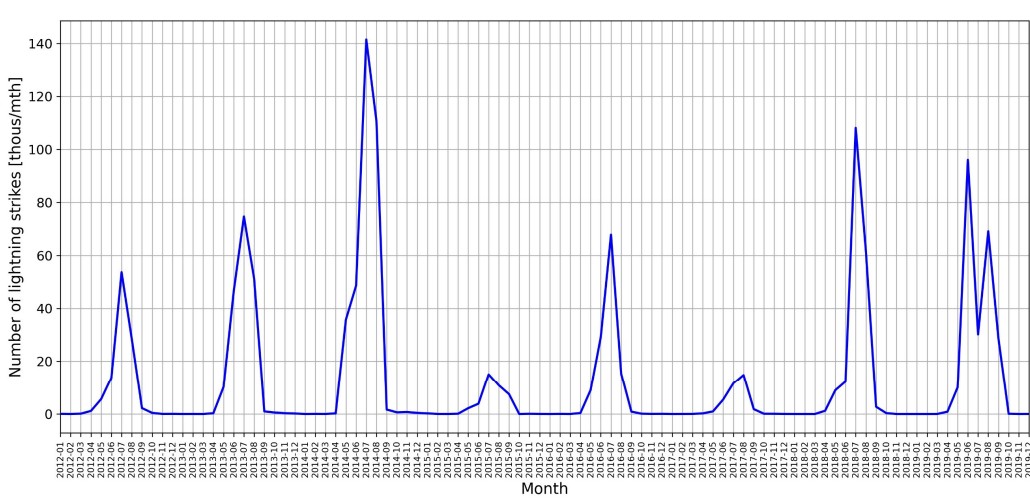

**Figure A1.** Monthly lightning strikes over Sweden between 2012 and 2019.

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
