# Peer review of "Regional Variability and Driving Forces behind Forest Fires in Sweden"

_remotesensing, doi:10.3390/rs14225826_

Round 1

Reviewer 1 Report

In this study, a machine learning method was developed to assess the probabilities of burned areas for Swedish forests. The analysis results about feature importance and their variability across Sweden provide valuable information to understand the vulnerability of Swedish forest to fire. In general, the topic can contribute to the forest fire prevention and management studies. However, there are some major flaws in the manuscript which need to be addressed for a possible revision.

First of all, the structure of the manuscript should be improved. The description of methods can be more concise and more informative findings of the study should be added in the abstract. The introduction section is too shallow in this version and should be rewritten. A thorough review of the forest fire modeling approaches should be given in the introduction section. The research gaps in this field and how the presented work can fill these gaps should also be clearly stated.

Secondly, the forest fire modeling approaches were introduced in the materials and methods section. This part should be moved to the Introduction section. The random forest fire concurrence model the authors developed should be described in detail in the materials and methods section. I also suggest the authors add a flow chart to illustrate the data processing workflow they used.

Thirdly, I suggest the authors add some policy implications on forest fire prevention considering the modeling and evaluation results the obtained in the discussion section. Typos and English editing should be checked throughout the manuscript.

There are some minor mistakes in the manuscript. For example:

Line 103: What does FFMC values mean? Can the authors give more explanation?

Line 328: “Modelling because of explanatory factor variability and model calibration dataset size caused extensive computer processing time.”?

Figure 1: The study area map should be redrawn. The geographic location of the study area is not clear at this stage.

Figure 3: The image is too blurry and the labels cannot be seen clearly.

Figure 4: What does the “thr” on X axis mean?

Author Response

Dear Editors,

Thank you for your time and comments. We improved the manuscript according to your suggestions and hopefully we can publish soon.

Best regards,
Reinis Cimdins

Reviewer 2 Report

The study “Regional Variability and Driving Forces Behind Forest Fires 2 in Sweden” uses valuable data and shows nice results. However, I have major concerns that should be addressed before publication in any international peer-reviewed journal:

1-     Formal aspects: the work should be strongly revised to  include the information of each section in the proper place (I mean, the introduction in the introduction, methods in methods, results in results…etc). Structureof some sections should be also improved. Currently, many information is explained in the unproper order and location, or mixed with the information corresponding to that section.

2-     Figures should be improved (eg. whats the meaning of thr and value in Fig. 4 axis?)

3-     RF model should be better described and results properly interpreted as collinear predictors are used to calculate variable importances.

Minor comments:

L41: Please, revise the citation format. I think should be [1-3]. Also revise the use of brackets in the rest of the document.

L47: To be honest, that statements should be softened. Although there is robust scientific evidence supporting that fire seasons are lengthening in many regions, burned area is decreasing worldwide (and thus fire frequency) (Andela et al., 2017). Likewise, no global analyses have been published yet supporting a global increase in burn severity.

L48-51: Please, improve the quality of writing

L55: I would like to see a more structured introduction before stating your objectives. (Just as a suggestion Eg. First paragraph about fire in the globe, another about fire drivers, and a last one focusing on Sweeden)

L59-84: This content is not properly placed. Please, rephrase/write in a different way to state before your objectives or, if those are your methods, move to methodology.

L95 A point is missing

L114: Please, resize figure 2 to be able to read labels without difficulties.

L101-104: Are these results of your analyses? If so, move to results or explain in a different way to fit that better in the methods section.

L119: Please, note that you have not talked about the fire occurrence model yet (which is explained in next sections).

L130: And lightning was not in that database? Why you have talked about lightning in the previous paragraph apart from the database? Maybe a first paragraph briefly providing structure to your data would be nice.

L173: This section is totally unnecessary and unproperly located. It mixes introduction, results and discussion of other researchers results. To briefly justify your approach in the methods section is enough.

L206: Here you talk about your statistical methods, more details such as model tuning (ntree, mtry) would be nice for allowing reproducibility of your study.

L214: Your validation approach would provide similar results to the Out-of-bag calculations in the RF model. It makes some sense but by this reason is somehow unnecessary. For a robust validation you should use independent regions (not points in your training regions, potentially strongly affected by spatial autocorrelation).

L252-254: This is methodology

L264-280: I like your way of explaining importance variables by groups of correlated variables. This is very important, as collinearity largely affect variable importances. In this sense, a poor non-collinear predictor can reach a higher importance value in a RF model than several collinear predictors that would perform much better than the former if individually. However, do you show predictor’s correlation in some place?

L278: You are mixing results and discussion. A clear example is L278-279 where you explain potential reasons of your results (discussion).

References

Andela N. et al., A human-driven decline in global burned area. Science 356, 1356–1362 (2017).

Author Response

(The authors gave the same response as above.)

Round 2

Reviewer 2 Report

Dear authors,

Thank you very much for your effort. I feel that the manuscript has been largely improved. I hope my comments were helpful.

Best,